# Role of Parity and Age in Cesarean Section Rate among Women: A Retrospective Cohort Study in Hail, Saudi Arabia

**DOI:** 10.3390/ijerph20021486

**Published:** 2023-01-13

**Authors:** Reem Falah Alshammari, Farida Habib Khan, Hend Mohammed Alkwai, Fahaad Alenazi, Khalid Farhan Alshammari, Ehab Kamal Ahmed Sogeir, Asma Batool, Ayesha Akbar Khalid

**Affiliations:** 1Department of Family and Community Medicine, College of Medicine, University of Ha’il, Ha’il 55476, Saudi Arabia; 2Department of Pediatrics, College of Medicine, University of Ha’il, Ha’il 55476, Saudi Arabia; 3Department of Pharmacology, College of Medicine, University of Ha’il, Ha’il 55476, Saudi Arabia; 4Department of Medicine, College of Medicine, University of Ha’il, Ha’il 55476, Saudi Arabia; 5Maternity and Child Hospital, Ha’il 55471, Saudi Arabia; 6William Harvey Hospital, East Kent Hospitals University NHS Foundation Trust Kent, Canterbury CT1 3NG, UK

**Keywords:** parity, age, BMI, mode of delivery, cesarean section, Saudi Arabia

## Abstract

In the context of the global increase in the rate of cesarean deliveries, with an associated higher morbidity and mortality, this study aimed to investigate the role of maternal age and parity in the cesarean section rate among women in the Hail Region of Saudi Arabia. This retrospective cohort study used data collected from the labor ward of the Maternity and Child Health Hospital, Hail, over a period of 8 months, forming a cohort of 500 women. Women were categorized into four different parity classes. The results revealed that there was no significant relationship between cesarean deliveries and maternal age (*p*-value, 0.07). There was no significant difference in the mode of delivery between the study’s parity cohort group. A significant increase in cesarean deliveries was noticed among obese women with a BMI between 35–39.9 (52.14%). This increase was even greater among those with a BMI above 40 (63.83%). Fetal distress, malpresentation and abruptio placenta were the most significant indications for CS among all age groups (*p*-value 0.000, 0.021, and 0.048, respectively). Conclusions: The number of cesarean deliveries has no association with parity or age. However, there was a statistically significant association with BMI, a perineal tear after previous vaginal delivery, and a history of diabetes mellitus and gestational diabetes. The most reported reasons for CS were fetal distress, malpresentation, and abruptio placenta among all age groups.

## 1. Introduction

Worldwide, cesarean section (CS) remains the most common surgical procedure performed on women of childbearing age. One in three American women undergo CS, and it is the leading cause of maternal mortality and morbidity in the US [1]. In recent decades, it has been a major public health concern worldwide, especially with the global increase in the rate of cesarean deliveries (CDs), which according to a a nationally representative data from 1990 to 2018, was 21% (based on the latest available data (2010–2018) from 154 countries covering 94.5% of the world’s live births) [2].

Several studies indicated important maternal factors that are likely to be associated with the rising rate of CS, such as educational status, income, preference, parity, and other indications including cephalopelvic disproportion, hypertensive disorders of pregnancy, antepartum hemorrhage, previous cesarean section, and fetal distress [3,4,5,6,7,8,9]. The increase in cesarean section rate is also associated with other several factors, such as advanced maternal age, multiple pregnancies, breech presentation, suspected low infant birth weight, private hospital status [10], and an increasing maternal BMI [11].

Maternal factors are strong determinants of birth outcomes. These factors may be genetic, environmental, or health factors. Parity is one of the most important factors affecting the health of mothers and fetuses [12]. Increasing age and parity are considered risk factors for adverse pregnancy outcomes, and an associated higher rate of CSs has been found in these studies [13]. 

High parity is considered one of the main public health problems. It is common in developing countries, particularly in Arab countries such as Saudi Arabia, where large families are the trend [14]. While ensuring healthy lives and promoting well-being for everyone of all ages, pregnancy without risk is one of the objectives of the Sustainable Development Goals (SDGs) [15], especially in developing countries where 99% of maternal and neonatal mortalities are recorded [16,17].

In the last three decades, there has been an increase in CS rates in both developed and developing countries [17]. This rise in CS rates places a burden on public health services, especially when CS is performed unnecessarily, leading to economic and service deficiencies and consequently poorer maternal and child health outcomes. Compared to vaginal delivery, CS is documented to be associated with a higher risk of infection, urinary tract infection, pain, headaches, anesthetic complications, maternal death, and postpartum depression among other symptoms [18,19,20,21].

Based on this issue, the researchers of this study found it necessary to investigate the role of parity and age in deciding the mode of delivery among women in Hail, Saudi Arabia. Additionally, the researchers explored other factors affecting the mode of delivery such as maternal BMI, level of education, income, and maternal medical and obstetrical history.

### The Aim of the Study Is as Follows

Investigate the role of maternal factors on the rate of cesarean section among women in the Hail Region.

## 2. Materials and Methods

### 2.1. Design

The research was carried out through a retrospective cohort study.

### 2.2. Setting

Data were collected from the labor ward of the Maternity and Child Health Hospital, Hail Region, Kingdom of Saudi Arabia.

### 2.3. Patient and Method

The sampling technique used was convenience sampling.

The sample size was 500, which was calculated by applying the following formula for a single proportion:n = Zα/2^2^ × p × (1 − p)/MOE^2^

Zα/2 is the critical value of the normal distribution at α/2 (for a confidence level of 95%, α is 0.05 and the critical value is 1.96), MOE is the margin of error, p is the sample proportion, and n denotes the sample size. 

Charts of study participants were reviewed for all pregnant women who were booked or unbooked and underwent delivery in the Maternity and Child Health Hospital, Hail, Kingdom of Saudi Arabia. 

Inclusion criteria included all pregnant Saudi women who delivered a single neonate at a gestation age of ≥28 weeks. Exclusion criteria included pregnant women with multiple gestations, pre-existing illnesses that could cause adverse outcomes during pregnancy such as renal and cardiac diseases, known diabetics, smokers, and those with hypertension before their first pregnancy

Sociodemographic data and information on pregnancy-related complications such as gestational diabetes, hypertension (de novo hypertension alone after 20 weeks of gestation in a previously normotensive woman), etc., as well as patient medical records and past obstetrical records, were retrieved from participants’ files. Participants were followed in the postnatal ward till their discharge.

Women were classified into 4 groups according to parity: primigravida (1 birth), multipara (2–4 births), grand multipara (5 or more births), and great-grand multipara (10 or more births).

A sample size of 457 ensures a maximum error of estimates of 5 percentage points at the 95 confidence level.

### 2.4. Statistical Analysis

Data were analyzed using SPSS software v.25.0 for Windows^®^ (SPSS Inc., Chicago, IL, USA). Differences between groups were assessed using the chi-square (χ^2^) test (Fisher’s exact test and likelihood ratio (LR)). *p* ≤ 0.05 was the significance threshold. In addition, logistic regression of the cesarean section rate was performed. Enter method was used for variable selection.

### 2.5. Ethical Consideration

Informed verbal consent at the time of admission was given by all participants prior to their participation in the study to use their medical records from their previous and current pregnancies.

The study was approved by the Research Ethics Standing Committee (REC) of the university of Ha’il (H-2022-010).

## 3. Results

The sample size was achieved in 8 months (from the 1 October 2021 to 31 May 2022).

Table 1 shows that the peak childbearing age was between 25 and 35 years of age (50% of total deliveries). Women with multiparity accounted for 47% of the total deliveries. Of the 500 deliveries that occurred during the study period, 243 cesarean sections were performed, with an overall incidence of 48.6%. No significant association was found between the CS rate and age with a *p*-value of 0.07. There was no significant difference in the mode of delivery between the study parity cohort groups, as shown in Table 1.

Table 2 shows that the majority were housewives (86.80%), with a monthly income of less than 15,000 SR (90.80%), and had an educational level that ranged from high school (43.00%) to intermediate school (34.00%). There was a significant relationship between BMI and the mode of delivery. There was a higher incidence of CS among women with a BMI between 35–39.9 and an even higher incidence among morbidly obese women (52.14% and 63.83%, respectively). 

Table 3 shows that fetal distress, malpresentation, and abruptio placenta were the most significant indications for CS among all age groups (*p*-value, 0.000, 0.021 and 0.048, respectively). Table 4 shows a statistically significant prevalence of some indications of CS such as failure of induction, malpresentation, abruptio placenta, and placenta previa among all parity groups (*p*-value, 0.000, 0.000, 0.035, 0.049, respectively). There were six women (not shown in the table) who underwent an elective cesarean section procedure based on their preference and request without having any medical indications.

Table 5 shows a significant relationship between maternal history of anemia, diabetes mellitus (DM), gestational diabetes (GDM) previous history of perineal tear, and history of pre-eclampsia with *p*-values of 0.005, 0.001, 0.01, 0.002, and 0.021, respectively.

Table 6 shows the logistic regression of the CS rate with a significant relationship between a high CS rate and a past history of perineal tear, BMI, and past medical history of DM or GDM, with *p*-values of 0.012, 0.015, 0.030, and 0.035, respectively (ORs, 7.55, 1.036, 5.43, and 2.82, respectively.)

## 4. Discussion

The overall CS rate of 48.6% for the study period compares favorably with rates reported in developed countries. Lower rates of cesarian sections have been reported in the United States (24.4%) and Denmark (20.4%) [22,23]. The rate of CS is trending upward in Saudi Arabia, especially when the current rate is compared with earlier studies [23]. The current rate is comparable to the rate in the United States [24]. The CS rate results in this study confirm the increase in the overall CS rate in Saudi Arabia. This was shown in a 10-year review of CD rate in Saudi Arabia carried out in 2009 by Ba’aqeel where the overall CD rate increased by 80.2%, from 10.6% in 1997 to 19.1% in 2006, especially in King Faisal Specialist Hospital in Riyadh, and in Hail [25]. Another study published in 2014 in Saudi Arabia showed that CS deliveries comprised 19.05% of the 22,595 deliveries from 2008 to 2011, which also revealed an increase in the rate of CS in Saudi Arabia [26].

The higher rate of cesarean section in Saudi Arabia may be accounted for based on advanced maternal age (women 35 years of age or older), which is associated with an increase in CS rates [24]. Advanced maternal age may be related to the tendency to postpone pregnancy due to the prolongation of life expectancy and the increase in educational and professional opportunities for women, in addition to the wide use of contraceptive methods and the availability of assisted reproduction [27,28,29]. This is observed despite the awareness of the higher rate of complications in a late pregnancy [30]. Another possible reason for the increasing CS rate is women’s preference to avoid the pain associated with a normal delivery; 55.35% of those who choose to undergo CS thought this method is associated with less pain than vaginal delivery [31].

This study did not show any association between the rate of CS and the age of the mother. Our results differ from the results shown by a study performed by Martinelli et al. (2021) that studied the association between parity and the mode of delivery in women of advanced maternal age. They observed that women of an advanced maternal age were more likely to undergo cesarean section [30]. Inconsistency between our results and the literature may be due to a smaller sample size in our study compared with the sample size of studies such as the one carried out by Qublan et al. where 7671 deliveries were included [13].

Our study’s results align with previous studies concerning the relationship between CS rate and BMI. A study conducted by Barau et al. found a linear association between maternal BMI and the risk for cesarean section, and the trends were similar; an increased BMI increased the risk of cesarean section [32]. Obese women were six times more likely than non-obese women to have a cesarean section due to cephalopelvic disproportion or a failure to progress [33].

In the analysis of the Consortium on Safe Labor data, Boyle et al. reported that the most common indications for cesarean delivery overall were failure to progress, non-reassuring fetal heart tracing, and fetal malpresentation [34]. Another study that was carried out in Saudi Arabia showed that breech presentation, failure to progress, and cephalopelvic disproportion were the main indications for primary cesarean section [35]. In our study, the common indications of CS among all age groups were fetal distress, malpresentation, and abruptio placenta. Failure of induction, malpresentation, abruptio placenta, and placenta previa were also significant among all parity groups.

The significant association between the percentage of CS and GDM and preexisting DM found in this study was also detected in a retrospective study conducted in the US in 2015. This association may represent an independent association between GDM and CS rate that could be explained by the higher incidence of fetal macrosomia and other indications for cesarean delivery, such as obesity and previous cesarean delivery, in patients with GDM [36].

### 4.1. Recommendation

The rise in the CD rate is alarming. Given the associated morbidity and mortality, it is imperative that a national strategy be considered to reduce at least some of the unnecessary CDs in Saudi Arabia. It is important to develop programs or guidelines to reduce C-section rates while maintaining maternal and neonatal safety.

### 4.2. Study Limitations

As the study represents pregnant women who were admitted to the labor ward of the Maternity and Child Health Hospital, Ha’il Region, Saudi Arabia, the findings may not be generalizable. The significant findings represent association, not causation, and should be interpreted with caution due to the nature of our study design (cross-sectional). Other limitations include the retrospective nature of our analysis.

## 5. Conclusions

This study concluded that no association was found between parity or age and the mode of delivery. However, the association between the rate of cesarean section and BMI was statistically significant, as was the association with a perineal tear after previous vaginal delivery, and a past medical history of DM and GDM. The most reported reason for CS was the failure of induction.

## Figures and Tables

**Table 1 ijerph-20-01486-t001:** Association of age and parity with type of delivery.

		Type of Delivery				Total (*n* = 500)		Likelihood Ratio	
		CS (*n* = 243) (48.6%)		NVD (*n* = 257) (51.4%)					
		*n*	%	*n*	%	*n*	%	Test Value	*p*-Value
Age group	<25	30	58.8%	21	41.2%	51	10.2%	5.327	0.07
	25–35	137	54.8%	113	45.2%	250	50.0%		
	35 or more	90	45.2%	109	54.8%	199	39.8%		
Parity	Primipara	82	33.7%	77	30.0%	159	31.8%	1.962	0.580
	Multipara	112	46.1%	123	47.9%	235	47.0%		
	Grand multipara	45	18.5%	55	21.4%	100	20.0%		
	Great-grand multipara	4	1.6%	2	0.8%	6	1.2%		

**Table 2 ijerph-20-01486-t002:** Association of socio-demographic characteristics with type of delivery.

		Type of Delivery				Total (*n* = 500)		Likelihood Ratio	
		CS (*n* = 243)		NVD (*n* = 257)					
		*n*	%	*n*	%	*n*	%	Test Value	*p*-Value
Occupation of mother	Housewife	207	47.70%	227	52.30%	434	86.80%	1.076	0.300
	Working	36	54.55%	30	45.45%	66	13.20%		
Income	>15,000	215	47.4%	239	52.6%	454	90.8%	3.116	0.211
	15,000–30,000	24	61.5%	15	38.5%	39	7.8%		
	>30,000	4	57.1%	3	42.9%	7	1.4%		
Level of Education	Illiterate	33	56.90%	25	43.10%	58	11.60%	7.937	0.160
	Elementary school	9	33.33%	18	66.67%	27	5.40%		
	Intermediate school	76	44.19%	96	55.81%	172	34.40%		
	High school	108	50.23%	107	49.77%	215	43.00%		
	Bachelor	13	65.00%	7	35.00%	20	4.00%		
	Higher education	4	50.00%	4	50.00%	8	1.60%		
BMI Classification	<18.5	4	100.00%	0	0.00%	4	0.80%	26.521	0.000 *
	18.5–24.9	19	54.29%	16	45.71%	35	7.00%		
	25–29.9	33	33.33%	66	66.67%	99	19.80%		
	30–34.9	66	43.71%	85	56.29%	151	30.20%		
	35–39.9	61	52.14%	56	47.86%	117	23.40%		
	≥40	60	63.83%	34	36.17%	94	18.80%		
Chi-square	X^2^	66.157							
	*p*-value	<0.001 *							

* = significant *p*-value (<0.05).

**Table 3 ijerph-20-01486-t003:** Correlation of different indications for cesarean section with age. (Keeping the level of significance *p* ≤ 0.05).

	Age Group			Total	*p*-Value
	Less than 25	25–35	>35		
Malpresentation	2	7	5	14	0.021
	6.9%	14.3%	4.0%		
Macrosomia	1	0	8	9	0.885
	3.4%	0.0%	13.5%		
Failure of induction	20	24	71	115	0.125
	69.0%	49.0%	72.3%		
Fetal Distress	2	4	3	9	0.000
	6.9%	8.2%	4.2%		
Failure of progress of 1st and 2nd stage of labor	0	2	18	20	0.196
	0.0%	4.1%	24.9%		
Previous 1 scar	0	2	6	8	0.959
	0.0%	4.1%	10.1%		
Previous 2 or more scars	0	3	30	33	0.959
	0.0%	6.1%	36.2%		
Placenta previa	0	2	5	7	0.143
	0.0%	4.1%	4.0%		
Abruptio placentae	2	3	6	11	0.048
	6.9%	6.1%	10.1%		

**Table 4 ijerph-20-01486-t004:** Correlation of different indications for cesarean section with parity. (Keeping the level of significance *p* ≤ 0.05).

Reason for LSCS	Parity				Total	*p*-Value
	Primipara	Multipara	Grand Multipara	Great-Grand Multipara		
Malpresentation	9	5	0	0	14	0.000
	12.3%	4.3%	0.0%	0.0%	5.8%	
Macrosomia	2	3	4	0	9	0.060
	2.7%	2.6%	7.7%	0.0%	3.7%	
Failure of induction	41	44	29	1	115	0.000
	56.2%	38.3%	55.8%	33.3%	47.3%	
Fetal Distress	7	2	0	0	9	0.050
	9.6%	1.7%	0.0%	0.0%	3.7%	
Failure of progress of 1st and 2nd stage of labor	0	15	5	0	20	0.300
	0.0%	13.0%	9.6%	0.0%	8.2%	
Previous 1 scar	1	6	1	0	8	0.670
	1.4%	5.2%	1.9%	0.0%	3.3%	
Previous 2 or more scars	2	27	3	1	33	0.050
	2.7%	23.5%	5.8%	33.3%	13.6%	
Placenta previa	0	4	3	0	7	0.049
	0.0%	3.5%	5.8%	0.0%	2.9%	
Abruptio placentae	3	6	1	1	11	0.035
	4.1%	5.2%	1.9%	33.3%	4.5%	

**Table 5 ijerph-20-01486-t005:** Association between past medical history and cesarean section rate.

PMH	Type of Delivery				Total		Fisher’s Exact Test
	CS		NVD				
	*n*	%	*n*	%	*n*	%	*p*-Value
Anemia	10	4.1%	27	10.5%	37	7.4%	0.005 *
Hypertension	8	3.3%	3	1.2%	11	2.2%	0.094
DM	14	5.8%	2	.8%	16	3.2%	0.001 *
GDM	22	9.1%	6	2.3%	28	5.6%	0.001 *
Cardiac Disease	4	1.6%	1	0.4%	5	1.0%	0.169
Perineal Tear	13	5.3%	2	0.8%	15	3.0%	0.002 *
APH	6	2.5%	2	0.8%	8	1.6%	0.125
Placenta previa	6	2.5%	1	0.4%	7	1.4%	0.053
preterm labour	21	8.6%	14	5.4%	35	7.0%	0.110
Pre-Eclampsia	11	4.5%	3	1.2%	14	2.8%	0.021 *

Footnote: * ( significant *p*-value (<0.05), PMH (past medical history), Dm (diabetes melitus), GDM (gestational diabetes mellitus), APH (antepartum hemorrhage).

**Table 6 ijerph-20-01486-t006:** Logistic regression analysis of cesarean section rate.

	*p*-Value	Odd Ratio	95% C.I. for Odd Ratio	
			Lower	Upper
BMI	0.015 *	1.036	1.007	1.065
Past Medical History of Anemia	0.004 *	0.304	0.134	0.691
Past Medical History of DM	0.030 *	5.435	1.181	25.016
Past Medical History of GDM	0.035 *	2.821	1.075	7.403
Past History of Perineal Tear	0.012 *	7.555	1.552	36.787

Footnote: * (significant *p*-value (<0.05)).

## Data Availability

The data presented in this study are available on request from the corresponding author.

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
