# Peer review of "Role of Parity and Age in Cesarean Section Rate among Women: A Retrospective Cohort Study in Hail, Saudi Arabia"

_ijerph, 2023, doi:10.3390/ijerph20021486_

Round 1

Reviewer 1 Report

Instead of women you better said charts of women

avoid some vague sentences like line 70

questioner replaced by check list

since you reviewed charts so from whom you got consent?

On discussion describe marriage age but not described on the result section

The age classification was not in a constant age range

Para 4 not sepecified The Cesarean section rate in the P0–>P3 parity groups 191 was 8.5, 7.1, 7.4, 6.3 and 12.9%

Author Response

The reply is attached as  Word file 

Reviewer 2 Report

Description of study site and study population is not detailed enough.

How sample size of 500 was chosen? And why 8 months duration for data collection was set? How the study participants were recruited?

Does the study involved all the deliveries meeting inclusion criteria during 8 months duration? Or, the delivery data of the hospital was randomly choosen?

Variable definitions are not that clear and uniform. Need to define variables like bio socio demographics characteristics, socio demographics data, maternal obstetric data and perinatal data.

Epi info is used for data entry or data analysis as well?

In line no 108, chisquare test was done to find differences between two groups is written. It is not clear. Analysis section should be much detailed. And another suggestion is to use logistic regression, to control various other confounders to show association between your major dependent and independent variables.

How age categories are made? Interval is not uniform. And below 18 category could be different than 18-25 category. One single category of below 25 could be misleading, thus suggest to refer to other literatures and recategorize age.

How you can say that the peak childbearing age is 30-40 in line no 117, it has 10 interval and other has 5, thus it is not an appropriate sentence.

Table 1 title, correlation word in not suitable since “correlation” can be used only for continuous variables.

In Table 2, what is logic behind categories of income, why less than 15000 is made one category and another with 5 interval only? Same interval would be more logical. And >30,000 catergory would be enough instead of having category of >40,000 with only 2 frequency. Chi square test with less than 5 cell frequency will need some correction, have you done it ?

Need operational definition for category of education. Uneducated, primary and middle is not clear. Does uneducated refer to never been or school or illiterate, some people are literate but they have never been to school. What is the difference between primary and middle is not clear.

In Table 2, I would prefer row percent instead of column percent.

What is the prevalence of various indications of cs is not clear, and also whether it is elective or indicative is also not clear.

In first paragraph of discussion, more recent publications on CD should be cited

References

1. Al Rowaily MA, Alsalem FA, Abolfotouh MA. Cesarean section in a high-parity community in Saudi Arabia: clinical indications and obstetric outcomes. BMC pregnancy and childbirth. 2014 Dec;14(1):1-0.

 2. Alabdullah HA, Ismael L, Alshehri LA, Alqahtani S, Alomari M, Alammar A, Ahamed SS. The Prevalence of C-Section Delivery and Its Associated Factors Among Saudi Women Attending Different Clinics of King Khalid University Hospital. Cureus. 2021 Jan 18;13(1).

Second paragraph of discussion need to be backed up by references, is there any evidence of increase age of pregnancy, and is there data to justify your claim for women performing cs to avoid pain?

Discussion section lacks the limitations of the study, it seem there are many limitations in this study, and limitation section is missing.

Author Response

The reply is attached as Word file 

Round 2

Reviewer 2 Report

Age categorization still need revision and Odds ratio calculation need to be revisited. Age categorization will greatly impact the association. I suggest less than 25, 25-35 and 35 and above categories as shown in other literatures. Above 35 is regarded as advanced maternal age.

Row percent should be shown in tables, please refer to other articles as well to present your findings in the table.

Author Response

done in the attached file 

Thank you 
